# Oral health inequality in Canada, the United States and United Kingdom

**Malini Chari** [1] *, **Vahid Ravaghi** [2], **Wael Sabbah** [3], **Noha Gomaa** [4‡], **Sonica Singhal** [1‡], **Carlos Quiñonez** [1]

**1** Faculty of Dentistry, University of Toronto, Toronto, Ontario, Canada, **2** School of Dentistry, University of Birmingham, Birmingham, England, United Kingdom, **3** Faculty of Dentistry, Oral & Craniofacial Sciences, King's College London, London, England, United Kingdom, **4** Schulich School of Medicine and Dentistry, Western University, London, Ontario, Canada

☉ These authors contributed equally to this work.
‡ NG and SS also contributed equally to this work.
* malini.chari@mail.utoronto.ca

**Data Availability Statement:** 1. All relevant data for the United States and United Kingdom are within the paper and its Supporting Information files. 2. The Canadian data underlying the results presented in the study are available from the

## Abstract

The objective of this study was to quantify the magnitude of absolute and relative oral health inequality in countries with similar socio-political environments, but differing oral health care systems such as Canada, the United States (US), and the United Kingdom (UK), in the first decade of the new millennium. Clinical oral health data were obtained from the Canadian Health Measures Survey 2007–2009, the National Health and Nutrition Examination Survey 2007–2008, and the Adult Dental Health Survey 2009, for Canada, the US and UK, respectively. The slope index of inequality (SII) and relative index of inequality (RII) were used to quantify absolute and relative inequality, respectively. There was significant oral health inequality in all three countries. Among dentate individuals, inequality in untreated decay was highest among Americans (SII:28.2; RII:4.7), followed by Canada (SII:21.0; RII:3.09) and lowest in the UK (SII:15.8; RII:1.75). Inequality for filled teeth was negligible in all three countries. For edentulism, inequality was highest in Canada (SII: 30.3; RII: 13.2), followed by the UK (SII: 10.2; RII: 11.5) and lowest in the US (SII: 10.3; and RII: 9.26). Lower oral health inequality in the UK speaks to the more equitable nature of its oral health care system, while a highly privatized dental care environment in Canada and the US may explain the higher inequality in these countries. However, despite an almost equal utilization of restorative dental care, there remained a higher concentration of unmet needs among the poor in all three countries.

## Introduction

Despite significant improvement in population health over time in Western nations, differences in health between population subgroups remain, whereby poor health is concentrated in socioeconomically marginalised groups [1]. This is observed for population oral health as well [2]. The persistence of health inequality is attributed to sociopolitical contexts, particularly to the role played by the welfare state [3]. Welfare states with more generous, universal and (re)

Research Data Centre in Toronto after receiving approval from Statistics Canada. The data can be accessed by others through a Data Liberation Initiative agreement, just as the authors did via an application process. The contract between authors and Statistics Canada has been uploaded as supplementary files Link for application: https://www.statcan.gc.ca/en/microdata/dli The contact for enquiries is statcan.mad-damdam-mad.statcan@canada.ca.

**Funding:** This research was supported through the generous funding of the Canadian Dental Protective Association and Green Shield Canada. The funders had no role in study design, data collection and analysis, decision to publish, or preparation of the manuscript.

**Competing interests:** I have read the journal's policy and the authors of this manuscript have the following competing interests: Dr. Carlos Quiñonez receives remuneration from Green Shield Canada for consulting services around dental care related issues. All the other authors declare no conflict of interest. This does not alter our adherence to PLOS ONE policies on sharing data and materials

distributive policies, and with broader population coverage and stronger resource allocation mechanisms for social benefits, have significantly lower health inequality [3, 4]. However, while country comparisons for population oral health reveal that liberal welfare states with market-dominated economies and more limited (re)distributive policies tend to have worse oral health, inequality in oral health is not always systematically greater [5].

Political and economic contexts serve as structural determinants of health outcomes, stratifying the population based on income, occupation, education, gender and ethnicity, which ultimately mediate inequality in health. Given this, it would seem reasonable to assume that socioeconomic inequality in oral health results from these contextual characteristics, which impact the distribution of relevant resources, particularly income. In turn, it would also seem reasonable that oral health inequality stems from the way these structural determinants modify more immediate determinants such as dental behaviors and the utilization of and access to dental services, reflecting the social, living and working conditions in which behavioral choices or decisions are made, ultimately shaping oral health inequality as well [6]. This suggests that differences in the organisation, financing and delivery of oral healthcare in a country may also provide a unique explanation for socioeconomic inequality in oral health, apart from broader political and social arrangements [5].

Canada, the United States (US) and the United Kingdom (UK) are liberal welfare states that have low public health expenditure and (re)distributive social spending, as well as higher income inequality than more egalitarian nations [7–10] (Table 1). Nevertheless, Canada, the US and the UK differ in terms of their healthcare and oral healthcare systems. Both Canada and the UK have a national system of universal health insurance covering hospital and physician care, yet Canada excludes oral healthcare and the UK does not [11] (Table 1). Despite the rise of private practice in the UK, the National Health Service (NHS) remains the dominant provider of dental care at subsidized rates to all citizens, with patients contributing about half of total oral healthcare expenditure [12]. While in Canada and the US, dental care is largely

**Table 1. Comparative framework to analyse oral health inequality in Canada, the United States and United Kingdom.**

|  | Canada | United States | United Kingdom |
|---|---|---|---|
|  | 2010 | 2010 | 2010 |
| Total healthcare expenditure[a] | 10.70 | 16.30 | 8.40 |
| Public healthcare expenditure[a] | 7.50 | 7.90 | 7.07 |
| Public social spending[a] | 17.5 | 19.3 | 22.4 |
| Income inequality[b] | 0.32 | 0.38 | 0.34 |
| Oral Healthcare System Features |  |  |  |
|  | Canada | United States | United Kingdom |
| Total oral health expenditure[c] | 6.0% | 4.2% | 4.0% |
| Financing of oral health care[d] | Private: 94% | Private: 89% | Private: 54% |
|  | Public: 6% | Public: 9% | Public: 46% |
| Population covered | Private: 62.6% | Private: 60.0% | Private: 11.8% |
|  | Public: 5.5% | Public: 5.0% | Public: 100% |
|  | No coverage: 32% | No coverage: 35% | No coverage: 0% |

[a] Expressed as percentage of GDP

[b] Gini Coefficient

[c] Expressed as percentage of total healthcare expenditure

[d] Expressed as percentage of oral healthcare expenditure

Adapted from: OECD 2021a Health spending (indicator), OECD 2021b Social spending (indicator), OECD 2021b Income inequality (indicator), Vujicic et al. 2016, and Boyle, 2011.

privatised, financed primarily by employment-based insurance, with limited contributions from government. Most dental care in Canada and the US is delivered in private settings on a fee-for-service basis, while the majority of care in the UK is delivered by the NHS in community and hospital settings under payment models that have changed over time, with some care delivered in private practices on a fee-for-service basis [11]. Ultimately, while it is known that sociopolitical context and the features of an oral healthcare system are bound to impact oral health and oral health inequality, no analysis has compared the magnitude of inequality for clinical oral health indicators within and between these three countries.

Although developing, the dental literature in this area remains limited. For example, Mejia et al. compared inequality in clinical and self-reported oral health among high income nations such as Australia, New Zealand, Canada and the US, and concluded that the availability of public dental services to low-income individuals mediated inequality in countries with the privatised delivery of oral health care [13]. Bhandari et al. highlighted the inverse association of public disinvestment in health care with dental service utilization [14]. Guarnizo-Herreño et al. are the only authors to explore oral health inequality between the US and the UK in terms of edentulism, missing teeth, self-reported oral health and oral impacts on daily life [15]. They found that both absolute and relative inequality were higher among Americans for both subjective and clinical outcomes. Their study speaks to the more equitable nature of the NHS, shedding light on how differences in the funding and delivery of dental care might impact the distribution of oral health in a country. Previous research comparing oral health inequality between Canada and the US has reported persistent inequality over time in both countries, with a disproportionate burden of oral disease concentrated among the poor, particularly in the US [16–18]. Despite highly privatised oral health care systems, inequality in the uptake of restorative services was found to decline in both countries, suggestive of enhanced access to dental services over time, which was however inadequate to address the burden of oral disease, particularly among the poor [18].

While there has been no comparative research on oral health inequality within and between Canada, the US and the UK, an assessment of how differences in socioeconomic status impact the distribution of oral health outcomes may provide important insights on the nature of oral health inequality in these countries, given their similar socio-political context and fundamental differences in the funding and delivery of oral health care. Therefore, this study aims to do just that by comparing both absolute and relative inequality across a variety of clinical indicators among adults, in Canada, the US and the UK in the first decade of the new millennium.

## Materials and methods

### Data sources

Data was obtained from nationally representative surveys in each country; the Canadian Health Measures Survey 2007–2009 (CHMS) in Canada, the National Health and Nutrition Examination Survey 2007–2008 (NHANES) in the US, and the Adult Dental Health Survey 2009 (ADHS) in the UK. These datasets were chosen as the CHMS and the ADHS were the most recent nationally representative datasets available in Canada and the UK, respectively.

The CHMS, conducted between March 2007 and 2009, collected clinical and demographic data from 5,586 Canadians age 6–79. People living in institutions, on crown land, in remote regions and members of the Canadian Armed Forces were excluded. A stratified multi-stage sampling technique was used, collecting information over two phases: demographic data via household interview, and oral health data via clinical examination. The combined response rate for the interview and examination was 51.7% [19].

The NHANES, conducted between January 2007 and December 2008, used a multi-stage probability sampling technique to collect data from 10,149 non-institutionalised Americans age 0–80. Data were collected over two phases via household interview followed by clinical examination with an unweighted response rate of 75.4% [20].

The ADHS, conducted between October 2009 and April 2010, collected data from 11,380 individuals age 16 and over, residing in England, Wales and Northern Ireland. A stratified multi-stage cluster sampling technique was used to collect data via a questionnaire-based household interview, followed by clinical oral examination of those having at least one natural tooth. The participation rates for the interview and examination were 84% and 61% respectively [21].

For this study, all individuals ≥16 years of age were included in the analysis of inequality in the three respective countries.

## Outcome variables

Our analysis focused on three clinical oral health indicators, based on the Decayed, Missing and Filled Teeth index (DMFT index). The first, ≥1 untreated decayed teeth (UD), wherein the D component of the DMFT index was used to indicate untreated decayed teeth, represents a measure of the burden of disease that remains unattended and estimates the level of unmet need in the sample population. It included pit and fissure, occlusal, proximal, overt and grossly decayed teeth that had never been restored to represent untreated decay levels in each population. The second, ≥1 filled teeth (FT), wherein the F component of the DMFT was used, represents the level of previous disease that had been treated and reflects the ability of individuals to procure treatment and the oral health system to deliver services. In other words, it represents the utilization of dental services. All permanent amalgam, composite resin and glass ionomer surface restorations along with previously filled teeth presenting with secondary decay and fractured/defective restorations were included in this variable. The third, edentulism or complete absence of teeth, represents unmet needs, utilization of dental services, and history of oral disease and behaviors over the life course. Individual tooth counts with assessment of each tooth surface was performed in the CHMS and ADHS to estimate both prevalence and severity of oral conditions, while a basic screening examination was carried out in the NHANES to assess prevalence. Edentulism was clinically verified in both the CHMS and NHANES, however, it was self-reported in the ADHS. To enable comparisons across surveys, all outcomes were dichotomised for analysis (Table 2).

## Socioeconomic status

Total annual household income and occupation-based social class were used as indicators of socioeconomic position in this analysis. The CHMS and NHANES reported total annual household income in an ordinal format ranging from 0 to >$100,000, which was ranked into

**Table 2. Categorization of outcome variables used in the analysis.**

| Variable | | Code | | Label |
|---|---|---|---|---|
| **Outcome Variables** | **Untreated Decay** | 0 | No decayed teeth present | No decay |
| | | 1 | ≥1 Decayed teeth present | Decay present |
| | **Filled teeth** | 0 | No filled teeth | No fillings |
| | | 1 | ≥1 filled teeth present | Fillings present |
| | **Edentulism** | 0 | 0–31 missing teeth | Dentate |
| | | 1 | 0–32 missing teeth | Edentate/Edentulous |

**Table 3. NS-SEC Classification of social class and annual household income as used in the analysis.**

| Social Class | Annual Household Income Groups |
|---|---|
| Managerial | Highest |
| Skilled non-manual | Higher middle |
| Skilled manual | Middle |
| Partly skilled | Lower middle |
| Unskilled | Lowest |

quintiles from highest to lowest. For the ADHS, social class was derived from the occupation of the respondent, based on the National Statistics Socioeconomic Classification (NS-SEC), which is derived from the household reference person's (HRP) occupational unit group and employment status, wherein the HRP is defined as the person responsible for owning or renting the accommodation. It was available as an ordinal variable in eight analytical categories; (i) higher managerial/professional, (ii) lower managerial/professional, (iii) intermediate, (iv) small employers and account workers, (v) lower supervisory and technical, (vi) semi-routine, (vii) routine, (viii) never worked and long term unemployed [22]. This was further collapsed into five categories to represent ~20% of the sample in each category (Table 3).

Importantly, educational attainment was not considered as the socioeconomic variable of choice due to inconsistency in the way it was captured in the three different surveys. While the Canadian and American surveys recorded the educational attainment of the respondent as a categorical variable ranging from high-school to post-secondary education, the UK ADHS only captured the age at which full-time education was completed. While the ADHS also provided information on the level of educational attainment, it was dichotomised to (i) at a degree-level or above and (ii) any other educational qualification. As a result, overall, the information was not comparable, nor did it provide a gradient to facilitate the use of the SII/RII.

Therefore, occupation-based social class was the next best choice for the socioeconomic variable after income, as it is widely used to measure socioeconomic gradients in the UK and has been previously used as a proxy measure in the absence of income-related data [22–24]. Moreover, it is derived from the HRP's occupational unit group and employment status, wherein the HRP is defined as the person responsible for owning or renting the accommodation. In the case of joint householders, the individual with highest income is taken as the HRP and in the case of householders with equal income, the oldest individual is considered the HRP. As a result, we found this variable to be more comparable to household income available in the CHMS and NHANES as opposed to educational attainment.

## Analysis

Inequality in health can be quantified by both simple and complex measures. However, complex indices such as the concentration index (CI), slope index of inequality (SII) and relative index of inequality (RII) are preferable, as they account for the hierarchical nature of socioeconomic position, which previously used indices such as the Gini index fail to account for [25]. As a result, complex measures of inequality reflect the experiences of the entire population, along with being sensitive to changes in the distribution of the population in each socioeconomic category [25, 26]. Such indices not only indicate the association of socioeconomic position to health, but also highlight how differences in socioeconomic status impact the distribution of health in a society. While the CI overcomes the limitation of the Gini index, the CI measures only relative inequality and is similar to the RII [25]. In our study we wanted to measure both absolute and relative inequality as they provide different information about the same population. Hence, we used the SII and RII,

two complex regression-based indices, to estimate absolute and relative inequality, respectively. While absolute inequality is the difference in health outcome between individuals at the highest and those at the lowest socioeconomic position and varies as the overall level of health in the population changes, relative inequality is the ratio or rate of change of health among those at the top of the socioeconomic ladder and those at the bottom, and informs where changes in health are occurring at the population level.

Both the SII and RII are estimated by the regression of the mid-point value of the health outcome to each socioeconomic group along a cumulative distribution. This is facilitated by the generation of a ridit score, which assigns values ranging from 0 to 1 to each hierarchically ranked socioeconomic group from highest to lowest, based on the midpoint of each socioeconomic category along the cumulative distribution. To determine the ridit score, weighted proportions of the socioeconomic variable (income and social class) were ranked from the highest level of income/social class to the lowest level assigning each category scores between 0 and 1, based on the mid-point of the cumulative distribution within each group. For example, if the highest income group consists of 20% of the population, its ridit score will be 0.1 (0.20/2), if the second highest group consists of 30% of the population, the ridit value will be 0.35 (0.2 +[0.3/2]) [26, 27]. The ridit scores were incorporated into linear regression models ($y = \alpha + \beta x$), testing for the association between each oral health outcome and the socioeconomic variable or the ridit score while adjusting for sex. The generated coefficient ($\beta$) is the SII or the absolute difference in the health outcomes, between the highest and lowest socioeconomic groups. The RII was obtained from the exponent of the regression coefficient β, which was achieved through the log-linear transformation of the dependent variable and is interpreted as a ratio. For this analysis, a positive SII value and an RII greater than 1 indicate inequality favouring the rich with higher concentration of unmet needs among the poor.

Procedures for complex sampling design on STATA version 15.0 were performed to conduct secondary data analysis. All individuals ≥16 years with complete data for all variables were included. This was done to maintain comparability and maximize the number of observations across the Canadian, US and UK datasets, as age in the ADHS 2009 dataset was reported as an ordinal variable that included 16-year-olds. The full sample comprised both dentate and edentulous individuals, while only dentate persons were included for the analysis of UD and FT. Age-standardised distributions of the outcomes across each socioeconomic group were estimated. Direct age-standardisation was performed to overcome the differences in the age composition in the three different samples. For this purpose, all three samples were mathematically adjusted to the standard population of US 2000 Census. In this way all three samples were given the same age distribution structure. The results therefore account for the differences in age composition across time and country [6, 28]. For each country, sex-adjusted estimates of inequality were generated using the SII and RII.

## Results and discussion

### Sample characteristics

The baseline characteristics of the sample population in all three countries are presented in Table 4. The age and sex distribution were similar. The prevalence of UD was highest in the UK (29.5%) and declined marginally as socioeconomic position increased (Table 4). Both Canada and the US showed similar socioeconomic gradients, with a much lower prevalence of UD in the highest socioeconomic categories than in the UK (Canada: 13%; US: 11.4%; UK: 23.2%). While the prevalence of FT was highest in Canada (89.2%), it was also relatively stable across socioeconomic categories in all three countries. The prevalence of edentulism was generally low across all the countries; however, it was marginally higher in the US (Table 5).

**Table 4. Sample characteristics.**

| | Canada | United States | United Kingdom |
|---|---|---|---|
| | 2007–2009 | 2007–2008 | 2009 |
| | n = 3981 | n = 5252 | n = 10130 |
| **Age[a,b]** | | | |
| 16–34 | 32.4 (29.6, 35.3) | 33.0 (31.06, 34.9) | 27.3 (24.6, 29.9) |
| 35–64 | 55.5 (53.5, 57.4) | 51.6 (49.9, 53.3) | 53.1 (51.4, 54.8) |
| ≥ 65 | 12.2 (10.1, 14.6) | 15.4 (13.7, 16.9) | 19.6 (17.5, 21.6) |
| **Sex[a,b]** | | | |
| Female | 50.6 (47.9, 53.3) | 51.2 (49.9, 52.5) | 51.3 (50.1, 52.5) |
| Male | 49.4 (46.7, 52.1) | 48.7 (47.4, 50.0) | 48.7 (47.4, 49.8) |
| **Socioeconomic Position[a,b,c]** | | | |
| Lowest | 24.5 (19.8, 29.8) | 22.3 (18.3, 26.2) | 13.1 (11.1, 15.0) |
| Lower middle | 18.7 (16.6, 21.1) | 20.4 (17.7, 23.1) | 28.2 (25.9, 31.7) |
| Middle | 16.6 (14.6, 18.8) | 15.7 (13.3, 18.1) | 21.5 (20.4, 22.7) |
| Higher middle | 12.4 (10.5, 14.7) | 20.4 (17.1, 23.7) | 24.0 (22.5, 25.5) |
| Highest | 27.8 (22.7, 33.5) | 21.1 (16.6, 25.5) | 12.5 (9.6, 15.4) |

[a] Weighted proportions expressed as percentage and 95% CI

[b] Based on full sample population

[c] Socioeconomic position for Canada and the US based on annual household income and for the UK based on social class

**Table 5. Age-standardized prevalence of oral health outcomes by socioeconomic position.**

| | | Canada | United States | United Kingdom |
|---|---|---|---|---|
| | | 2007–2009 | 2007–2008 | 2009 |
| **Presence of 1 ≥ Untreated Decay[a,b]** | | **20.3 (15.4, 26.3)** | **20.6 (17.7, 23.6)** | **29.5 (25.5, 33.5)** |
| **Socioeconomic[c] Position** | Lowest | 31.0 (25.0, 38.0) | 36.2 (30.4, 42.1) | 38.7 (33.0, 44.4) |
| | Lower middle | 24.0 (18.0, 32.0) | 26.3 (21.6, 31.0) | 31.8 (27.0, 36.5) |
| | Middle | 18.0 (11.0, 29.0) | 16.8 (13.3, 20.3) | 28.1 (22.8, 33.5) |
| | Higher middle | 17.0 (11.0, 27.0) | 15.8 (11.4, 20.1) | 26.9 (21.7, 32.2) |
| | Highest | 13.0 (8.0, 22.0) | 11.4 (8.02, 14.9) | 23.2 (17.4, 29.1) |
| **Presence of 1 ≥ Fillings[a,b]** | | **89.2 (86.3, 91.5)** | **81.5 (79.3, 83.7)** | **84.4 (82.3, 86.4)** |
| **Socioeconomic[c] Position** | Lowest | 85.0 (77.6, 90.3) | 81.1 (77.7, 84.6) | 81.1 (77.7, 84.6) |
| | Lower middle | 87.3 (82.3, 91.0) | 82.9 (79.5, 86.2) | 82.9 (79.5, 86.2) |
| | Middle | 91.4 (86.0, 94.8) | 85.7 (81.7, 89.8) | 85.7 (81.7, 89.8) |
| | Higher middle | 90.0 (85.0, 94.0) | 86.7 (81.7, 91.8) | 86.7 (81.7, 91.8) |
| | Highest | 91.0 (86.7, 94.1) | 84.2 (78.5, 89.8) | 84.2 (78.5, 89.8) |
| **Presence of Edentulism[a]** | | **3.2 (2.1, 4.8)** | **5.5 (4.2, 6.9)** | **4.2 (3.4, 5.1)** |
| **Socioeconomic[c] Position** | Lowest | 6.9 (4.1, 11.4) | 9.8 (6.8, 12.8) | 8.7 (6.7, 10.7) |
| | Lower middle | 3.3 (1.7, 6.2) | 5.8 (4.6, 6.9) | 5.7 (4.7, 6.8) |
| | Middle | 3.0 (1.4, 6.3) | 4.3 (2.03, 6.6) | 2.8 (2.03, 3.6) |
| | Higher middle | 2.9 (1.1, 7.5) | 3.07 (1.1, 4.9) | 2.3 (1.7, 3.02) |
| | Highest | 1.0 (0.4, 4.6) | 3.7 (0.65, 6.8) | 1.1 (0.5, 1.8) |

[a] Weighted proportions and 95% CI

[b] Decayed and filled teeth outcomes based on dentate population

[c] Socioeconomic position for Canada and the US based on annual household income and for the UK based on social class

## Inequality in oral health

Table 6 illustrates inequality in oral health outcomes within each respective country. There was significant absolute and relative inequality in UD in all three countries, which was lowest in the UK and highest in the US. The absolute inequality for UD in the US (SII: 28.2; 95% CI 21.9, 34.4) was almost double that of the UK (SII: 15.8; 95% CI 9.5, 22.1), and in Canada (SII: 21.0; 95% CI 14.5, 27.6) it was less than the US but more than the UK. In comparison to the UK (RII: 1.75; 95% CI 1.34, 2.28), the relative inequality for UD in Canada (3.09; 95% CI 1.69–5.65) was almost double, and more than double in the US (4.7; 95% CI 3.0, 7.15).

Absolute and relative inequality for FT was small in comparison to other outcomes. The magnitude of absolute inequality for FT was small and insignificant in the UK (SII: –8.4; 95% CI –17.2, 0.12), while in Canada (SII: –8.4; 95% CI –14.1, –2.7) it was similar but significant. The largest absolute inequality was in the US (SII: –17.7; 95% CI –22.8, –12.6). The magnitude of relative inequality was almost negligible in both Canada and the UK, and marginally higher in the US (Table 6).

Absolute and relative inequality in edentulism was highest in Canada and lowest in the US (Table 6). The absolute inequality in edentulism in Canada (SII: 30.3; 95% CI 24.0, 36.7) was three times as high as that of the UK (SII: 10.2; 95% CI 8.03, 12.3) and the US (SII: 10.3; 95% CI 6.9, 13.7). The relative inequality was also highest in Canada (RII: 13.2; 95% CI 4.7, 36.7), followed by the UK (RII: 11.5; 95% CI 7.5, 17.5) and lowest in the US (RII: 9.2; 95% CI 3.7, 22.8).

## Discussion

Our results show a disproportionate concentration of adverse oral health outcomes, such as UD and edentulism, among socioeconomically disadvantaged groups in all three countries, wherein the proportion of unmet needs was considerably higher among the poor in the US and Canada, than among those in the UK. Overall, while the average burden of oral disease

**Table 6. Sex adjusted absolute and relative oral health inequality in Canada, United States and United Kingdom.**

|  |  | Absolute Inequality | Relative Inequality |
|---|---|---|---|
| **1 ≥ Untreated Decay[a]** |  |  |  |
| **Canada** | **2007–2009** | 21.0***(14.5, 27.6) | 3.09**(1.69, 5.65) |
| **United States** | **2007–2008** | 28.2***(21.9, 34.4) | 4.70***(3.08, 7.15) |
| **United Kingdom** | **2009** | 15.8***(9.5, 22.1) | 1.75**(1.34, 2.28) |
| **1 ≥ Fillings[a]** |  |  |  |
| **Canada** | **2007–2009** | -8.4**(-14.1, -2.7) | 0.91*(0.85, 0.97) |
| **United States** | **2007–2008** | -17.7***(-22.8, -12.6) | 0.81***(0.76, 0.86) |
| **United Kingdom** | **2009** | -8.5[NS](-17.2, 0.12) | 0.90*(0.82, 0.99) |
| **Edentulism[b]** |  |  |  |
| **Canada** | **2007–2009** | 30.3***(24.0, 36.7) | 13.2***(4.7, 36.7) |
| **United States** | **2007–2008** | 10.3***(6.9, 13.7) | 9.3***(3.7, 22.8) |
| **United Kingdom** | **2009** | 10.2***(8.03, 12.3) | 11.5***(7.5, 17.5) |

[a] Decayed and filled teeth estimates based on dentate population in sample

[b] Estimates for Edentulism based on whole population in sample

***p<0.001

**p<0.01

*p<0.05

[NS]Insignificant

(UD and edentulism) was worse in the UK, oral health inequality was worst in the US with the exception of edentulism, where Canada appears to perform poorly.

While sociopolitical contexts and their associated institutions and policies are considered the structural determinants of inequality [3, 4], the limited research on country comparisons of oral health inequality suggest that the role played by the oral healthcare system in mediating such differences may be more important [5, 6, 13]. The lack of public dental services has been highlighted as the key mediating factor [13, 14] in the exacerbation of oral health inequality. The differences in the extent and coverage of public dental services in these countries and associated barriers in access could potentially explain our findings. For example, in the UK, the NHS offers universal dental coverage at subsidized rates along with treatment at no cost to certain vulnerable groups such as children, pregnant women, and welfare-assistance recipient [11, 12], which may explain the low level of inequality overall. Only 5% of Americans have public dental coverage, with over a third of the population having no access to dental insurance [11, 29], which may explain why the US has the worse inequality overall. In Canada, despite a largely privatised oral healthcare system, there is availability of provincial programs to social assistance recipients, their dependents, and some seniors [30], which may explain lower inequality in comparison to the US. Finally, while the NHS in the UK offers some form of public dental coverage to every citizen, a third of the population in both Canada and the US do not have any form of dental insurance [11, 29–31], thus creating barriers in access to dental care among low-income groups, which may explain the higher concentration of unmet needs (i.e. UD) among the poor in the US and Canada.

Contrarily, inequality in edentulism was found to be highest in Canada, followed by the UK and lowest in the US. Previous research on inequality trends in total tooth loss in the UK show that improvements in edentulism were almost double in the highest social class than in the lowest [32], which is also reflected in the high relative inequality for this outcome in our results. One of the reasons for this could be the relatively lower prevalence of edentulism within a small cohort of individuals, producing high relative differences, as per the theory of "mathematical artefact" when explaining health inequality [33]. However, it must be taken into account that edentulism in UK was self-reported, which may have underestimated its prevalence, leading to a high relative inequality for this outcome.

Previous literature comparing inequality in edentulism between high income countries such as Canada and the US have identified increasing age and low income as the largest contributor to inequality along with lack of education [16, 17]. This is in line with the results of this study, as the rate of improvement in edentulism was lower for the poor, given the high RII values and the low SII. Our results also indicate higher utilization of dental services in the UK and Canada, and the higher rates of inequality could be attributed to various factors, such as culture or clinical decision-making over the life course. While an anthropological perspective on tooth loss over the life-course may thus be a plausible explanation, it doesn't necessarily explain differences in edentulism based on the available literature [34].

Our results also reflected a closing gap between the rich and poor for FT outcomes in all three countries, suggesting that more low-income individuals can procure restorative dental care. This is in contrast to previous research suggesting that a lack of public sector investment results in low utilization of dental care [14]. Despite higher utilization of restorative care among the poor, there was still a high concentration of unmet needs and edentulism among this population in all three countries, suggesting the potential effects of persistent inequality in these societies more generally, and in their oral healthcare systems more specifically.

The focus of this study was to compare inequality in clinical indicators of oral health and is consistent with previous research on inequality in both normative and subjective oral health measures among adults across high income countries such as Canada, the US, Australia and

New Zealand [13]. In this research, inequality is attributed to the oral health care system for clinical indicators, and for subjective indicators, inequality is attributed to psychosocial factors and the manner in which the societies deal with psychosocial stress [13]. Based on another study, oral health inequality in subjective indicators and normative measures did not appear to be systematically different from each other [5]. Nevertheless, a comparison between the US and UK reported inequality to be worse in US for pain, difficulty in eating, speaking etc., a finding that corroborates the results of our study [15].

It is noteworthy that despite having a comprehensive, universal and integrated healthcare system, and an almost equal utilization of dental services in the UK, the burden of oral disease remains higher in comparison to Canada and the US. In other words, greater utilization of restorative care was not found to be adequate in mitigating inequality in unmet needs, or in lowering the overall prevalence of UD. In this regard, it should be acknowledged that the nature of dental caries is one that is likely modified by behaviours. Yet, while it is known that those at the higher end of the social gradient are more likely to adopt new behaviours that improve oral health [35, 36], living and social conditions are significant determinants of behavioural choices [36, 37]. It must therefore be kept in view that while the oral healthcare system is a factor in mediating such differences, tackling oral health inequality necessitates addressing the sociopolitical aspects of the welfare state as well [5].

Our findings need to be interpreted with respect to certain strengths and limitations. To the best of our knowledge, this is the first study to compare the magnitude of inequality in Canada, the US and the UK, using clinical oral health data from nationally representative surveys. Additionally, we used robust and rigorous measures of inequality in line with recommendations from the WHO [26]. However, our findings are hypothesis-generating, and the possible explanations to our findings require further investigation. Another limitation is the inconsistent use of socioeconomic measures between Canada and the US versus the UK, in that the latter uses an occupation-based social class measure of socioeconomic position, which despite being widely used to describe socioeconomic gradients in the UK [22, 38], its direct comparability to household income is a limitation. However, occupation-based socioeconomic data has previously been used as a proxy measure in the absence of income-related data in research based on social stratification [23]. Moreover, the NS-SEC classification of social class used in this study is known to have a significant and distinctive relationship with income, wherein each category of occupation correlates to the monetary earnings associated with it and follows a clear income gradient, thereby facilitating its use as a proxy measure [22–24]. Finally, we did not control for education or employment status in the relationship between income/social class and oral health outcomes.

While the results of this study coincide with previous research demonstrating oral health inequality to be worse in the US than in the UK [15] and Canada [16–18], we found the prevalence of oral disease was higher in the UK. Although we did not empirically assess the role played by the oral healthcare system in mediating inequality, our findings suggest that the funding and delivery of oral healthcare may potentially contribute to an exacerbation of inequality and warrants future research. Other research opportunities include comparing the inequality between oral and general health in these countries to better understand the role played by health institutions in mediating both outcomes. Extending this research into exploring inequality patterns in children will also augment the understanding of the extent to which sociopolitical and health institutions may mediate inequality differences by age.

## Conclusion

There was significant inequality in adverse oral health outcomes in all three countries, but it was lowest in the UK, which may be attributed to the more equitable nature of the NHS than

the manner in which oral healthcare is provided to populations in Canada and the US. While higher inequality in Canada and the US may be attributed to a predominantly privatized funding and delivery model for dental services, the nature of the oral healthcare system alone still did not appear to reduce the overall burden of unmet needs in all countries. Although our findings suggest a significant role for the oral healthcare system in mediating inequality, it may still need to be viewed in the larger context of the welfare state and its ability to address the social determinants of oral health.

## Supporting information

**S1 File.**
(PDF)

**S1 Data.**
(DTA)

**S2 Data.**
(DTA)

## Author Contributions

**Conceptualization:** Malini Chari, Vahid Ravaghi, Wael Sabbah, Noha Gomaa, Sonica Singhal, Carlos Quiñonez.

**Data curation:** Malini Chari, Vahid Ravaghi, Wael Sabbah, Carlos Quiñonez.

**Formal analysis:** Malini Chari.

**Methodology:** Vahid Ravaghi.

**Supervision:** Vahid Ravaghi, Wael Sabbah, Noha Gomaa, Sonica Singhal, Carlos Quiñonez.

**Visualization:** Malini Chari, Carlos Quiñonez.

**Writing – original draft:** Malini Chari.

**Writing – review & editing:** Vahid Ravaghi, Wael Sabbah, Noha Gomaa, Sonica Singhal, Carlos Quiñonez.

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
