## [Decision Letter · Decision Letter 0]

10 Jan 2022

PONE-D-21-31005Oral Health Inequality in Canada, the United States and United KingdomPLOS ONE

Dear Dr. Chari,

Thank you for submitting your manuscript to PLOS ONE. After careful consideration, we feel that it has merit but does not fully meet PLOS ONE’s publication criteria as it currently stands. Therefore, we invite you to submit a revised version of the manuscript that addresses the points raised during the review process.

We look forward to receiving your revised manuscript.

Kind regards,

Frédéric Denis, Ph.D.

Academic Editor

PLOS ONE

Journal Requirements:

This research was supported through the generous funding of the Canadian Dental Protective Association and Green Shield Canada

This research was supported through the generous funding of the Canadian Dental Protective Association and Green Shield Canada. The funders had no role in study design, data collection and analysis, decision to publish, or preparation of the manuscript.

I have read the journal's policy and the authors of this manuscript have the following competing interests: Dr. Carlos Quiñonez receives remuneration from Green Shield Canada for consulting services around dental care related issues.  

All the other authors declare no conflict of interest.

Reviewers' comments:

Reviewer's Responses to Questions

**Comments to the Author**

1. Is the manuscript technically sound, and do the data support the conclusions?

Reviewer #1: Partly

Reviewer #2: Partly

2. Has the statistical analysis been performed appropriately and rigorously? 

Reviewer #1: I Don't Know

Reviewer #2: I Don't Know

3. Have the authors made all data underlying the findings in their manuscript fully available?

Reviewer #1: No

Reviewer #2: Yes

4. Is the manuscript presented in an intelligible fashion and written in standard English?

Reviewer #1: Yes

Reviewer #2: Yes

5. Review Comments to the Author

Reviewer #1: Dear Authors

This study mainly aimed to calculate and compare socioeconomic inequalities in oral health across three countries with the similar sociopolitical structure but different oral healthcare system including Canada, US and UK. And, two complex measures for inequality (i.e., absolute measure of Slop Index, and its related Relative Index of Inequality) were used to calculate inequalities. The study focused on three dichotomous indicators: 1) the untreated decayed teeth as a measure for the level of unmet needs (an adverse oral health outcome); 2) the filled teeth as a measure for the utilization of dental services need; and 3) edentulism representing unmet needs, utilization of dental services, and history of oral disease and behaviors over the life course (an adverse oral health outcome). Data came from three nationally representative surveys: CHMS (conducted in Canada, 2007-2009), NHANES (conducted in the US, 2007-2008), and ADHS (conducted in the UK, 2009). It is recommended a significant inequality in adverse oral health outcomes in all three countries, but it was lowest in the UK.

It is an important study. There are some minor issues, but I have an important concerns about the methodological elements of this study which I prefer to focus on it.

- The definition of outcome variables is not clear. The authors have not provided the precious detail about how to dichotomized “untreated decayed teeth”,” filled teeth”, and “edentulism”. The authors need to clarify how exactly the initial data separated into two possible values, maybe 0 and 1. Now it is not possible to correctly interpret the estimations for SI & RII. For example, consider the “untreated decayed teeth“ as the first indicator. If the dichotomous variable defined 0 for those had not any untreated decayed teeth, and 1 for the others, this variable would be an adverse outcome. As a result, positive SIs and RII greater than 1 indicates that untreated decayed teeth concentrated more among the rich. In the other word, the rich needs more help to satisfy needs for oral health. The interpretation of positive SIs and RII greater than 1 would be completely opposite if the If the dichotomous variable defined 1 for those had not any untreated decayed teeth, and 0 for the others. Now the authors just mentioned the words: “�1 untreated decayed teeth (UD)”, and “�1 filled teeth (FT)”. The dichotomizing approach for edentulism is also unclear.

Minor points:

- Existing literate on health inequality monitoring, different complex measures of inequality including both absolute and relative measures provided. Some of these measures have been used more in available publications such as Gini index and Concentration index (). It would be useful that authors explain why they decided to calculate oral health inequality based on two selected measures.

- The sample of the study included three surveys conducted before 2010 which seem to be old. The authors needs to explain the reasons or limitations they had for choosing old datasets.

- The authors mentioned that edentulism was clinically verified in both the CHMS and NHANES but was self-reported in ADHS. It is strongly suggested that authors mentioned the related limitation in the interpretation of results in the “Discussion”.

- The reason for selection of three countries of Canada, The US, and the UK needs more clarification. There are a couple of developed countries with the similar sociopolitical context, and various oral healthcare systems which have the chance included in the sample study of this paper.

Reviewer #2: Thank you for the opportunity to review this manuscript. It builds nicely on work which you published last year comparing the magnitude of oral health inequality over time in Canada and the United States. Whilst my comments are quite numerous, I hope that they will assist in strengthening presentation of your results.

General - I suspect that this manuscript was prepared at the same time as your J Public Health Dent and that this current manuscript now needs updating to cite your newly published research.

Abstract - Please review the first sentence - starting 'there is little evidence' is not the strongest start to draw in the reader. And you present evidence in your introduction that there are at least two other papers in addition to your newly published paper on this subject.

Introduction - Towards the end of the introduction, there is a brief sentence telling us that there is other research comparing oral health inequality between Canada and the US (with references 16 and 17 cited). The reader wants to know what those papers (and your newly published one) have already told us about this subject - not just that they exist.

Table 1 - Expenditure: Please ensure that you present figures in the same format (ie to either one or two decimal places). Also delete the words 'predominantly private financing and public/private mixed model' - these are superfluous - the figures show the nature of the private/public mix in each country.

Table 1 - Population covered: Needs a foot note with the source of these figures. Please also review the statement that 100% of UK public is covered by publicly provided dental services. Only around 50% of the UK population access NHS dentistry each year, which may be important to your discussion section when exploring why UK has the worst rate of untreated decay. Also consider including a figure for 'no coverage' in the UK.

Materials and methods

Outcome variables: Needs the term DMFT explained early in the section. Please clarify (for readers who may not be dental professionals) that edentulism means all teeth are missing.

Analysis - early on in this section, you need to state that only data relating to adults (aged over 16 years) was included in the analysis - it currently appears rather late in this section. You might even wish to state this in the data sources section.

SII ad RII should have a reference to assist the reader who wants to understand more about this type of analysis

Why and how did you age-standardise and sex-standardise the results - it wasn't clear why this was undertaken and how it aids interpretation of the data.

Results - Table 4 The format of the title used in Table 2 of your J Public Health Dent paper is clearer. Also the foot notes a and c do not seems to have a reference in the table itself

.

Table 5 - needs meaningful titles needed SII and RII need to be replaced by terms which can be understood by the reader without reference to the main text

Discussion - sentence 2 - where has the term 'average level of oral health' been defined and explore in the results? This would be clearer couched as 'Untreated decay was worst in the UK........etc.' Do your data provide evidence that the publicly funded care provided in UK is of a poorer standard of care than is available to US and Canadians? Or is there a difference of philosophy in relation to minimum-intervention dentistry in the UK compared to the other nations eg enamel-only caries is treated more conservatively rather than with fillings?

Why refer to no cost treatment for children in the UK when the paper is about adult oral healthcare? Are you suggesting that better care as a child has an impact on adult oral health?

Line 370 - where did Australia and New Zealand suddenly come from? Needs new references if you are leaving this sentence in the manuscript.

Line 377-387 - as highlighted in the first and last paragraphs of your discussion section, this is one of the most important findings of your study and should appear much earlier in the discussion section.

6. PLOS authors have the option to publish the peer review history of their article (what does this mean?). If published, this will include your full peer review and any attached files.

Reviewer #1: No

Reviewer #2: **Yes: **Wendy Thompson

---

## [Author Response · Author response to Decision Letter 0]

18 Mar 2022

Thank you for your review. All review comments have been addressed and uploaded in the file "Response to Reviewers"

---

## [Decision Letter · Decision Letter 1]

21 Apr 2022

Oral health inequality in Canada, the United States and United Kingdom

PONE-D-21-31005R1

Dear Dr. Chari,

We’re pleased to inform you that your manuscript has been judged scientifically suitable for publication and will be formally accepted for publication once it meets all outstanding technical requirements.

Kind regards,

Frédéric Denis, Ph.D.

Academic Editor

PLOS ONE

Additional Editor Comments (optional):

Reviewers' comments:

Reviewer's Responses to Questions

**Comments to the Author**

1. If the authors have adequately addressed your comments raised in a previous round of review and you feel that this manuscript is now acceptable for publication, you may indicate that here to bypass the “Comments to the Author” section, enter your conflict of interest statement in the “Confidential to Editor” section, and submit your "Accept" recommendation.

Reviewer #1: All comments have been addressed

Reviewer #2: (No Response)

2. Is the manuscript technically sound, and do the data support the conclusions?

Reviewer #1: Yes

Reviewer #2: Partly

3. Has the statistical analysis been performed appropriately and rigorously? 

Reviewer #1: Yes

Reviewer #2: I Don't Know

4. Have the authors made all data underlying the findings in their manuscript fully available?

Reviewer #1: Yes

Reviewer #2: Yes

5. Is the manuscript presented in an intelligible fashion and written in standard English?

Reviewer #1: Yes

Reviewer #2: No

6. Review Comments to the Author

Reviewer #1: Dear authors

Thank you for submission of the revised version of your manuscript titled "Oral health inequality in Canada, the United States and United Kingdom". The authors did a very good job of editing the paper and addressed each specific comment. I have not any new comment.

Reviewer #2: Sorry - your responses have made me more uncomfortable with the paper than I was when first reading it.

7. PLOS authors have the option to publish the peer review history of their article (what does this mean?). If published, this will include your full peer review and any attached files.

Reviewer #1: No

Reviewer #2: No

---

## [Editor Report · Acceptance letter]

25 Apr 2022

PONE-D-21-31005R1 

Oral health inequality in Canada, the United States and United Kingdom 

Dear Dr. Chari:

I'm pleased to inform you that your manuscript has been deemed suitable for publication in PLOS ONE. Congratulations! Your manuscript is now with our production department. 

Kind regards, 

on behalf of

Dr. Frédéric Denis 

Academic Editor

PLOS ONE